# Diastereoselective Synthesis of (–)-6,7-Dimethoxy-1,2,3,4-tetrahydroisoquinoline-1-carboxylic Acid via Morpholinone Derivatives

**DOI:** 10.3390/molecules28073200

**Published:** 2023-04-04

**Authors:** Maria Chrzanowska, Agnieszka Grajewska, Maria D. Rozwadowska

**Affiliations:** Faculty of Chemistry, Adam Mickiewicz University, ul. Uniwersytetu Poznańskiego 8, 61-614 Poznań, Poland

**Keywords:** 1,2,3,4-tetrahydroisoquinoline-1-carboxylic acid, Petasis reaction, Pomeranz–Fritsch–Bobbitt cyclization, 3,5-substituted 1,4-oxazin-2-one

## Abstract

A simple and convenient synthesis of (–)-6,7-dimethoxy-1,2,3,4-tetrahydroisoquinoline-1-carboxylic acid is described, applying a combination of two synthetic methods: the Petasis reaction and Pomeranz–Fritsch–Bobbitt cyclization. The diastereomeric morpholinone derivative *N*-(2,2-diethoxyethyl)-3-(3,4-dimethoxyphenyl)-5-phenyl-1,4-oxazin-2-one formed in the Petasis reaction was further transformed into 1,2,3,4-tetrahydroisoquinoline-1-carboxylic acid via Pomeranz–Fritsch–Bobbitt cyclization, a classical method of synthesis leading to the tetrahydroisoquinoline core. We review important examples of applications of the Pomeranz–Fritsch process and its modifications in the synthesis of chiral tetrahydroisoquinoline derivatives that have been published in the past two decades.

## 1. Introduction

Isoquinolines and 1,2,3,4-tetrahydroisoquinolines (THIQs) are widely distributed in nature as alkaloids, and due to their diverse broad-spectrum biological activity are employed in medicinal chemistry [1,2,3]. THIQs containing natural and synthetic molecules have been qualified as ‘privileged scaffolds’ to identify, design, and synthesize novel biologically active derivatives of interest in drug development, from anti-inflammatory, anti-viral, anti-fungal, or anti-cancer compounds to Parkinson’s disease treatment [4,5,6,7]. 1,2,3,4-Tetrahydroisoquinoline carboxylic acids in their optically pure form are important building blocks for the synthesis of natural products and synthetic pharmaceuticals, and are currently subject to growing interest [8,9,10,11,12]. Recently, a series of optically active substituted 6,7-dihydroxy-1,2,3,4-tetrahydroisoquinoline-3-carboxylic acid derivatives (**A**) (Figure 1) such as esters and amides have been synthesized and evaluated as potent influenza virus polymeraze acidic (PA) endonuclease domain inhibitors [10]. 6,7-Dihydroxy-1-methyl-1,2,3,4-tetrahydroisoquinoline-1,3-dicarboxylic acid (**B**) (Figure 1), isolated from *Mucuna pruriens,* traditionally used for the treatment of Parkinson’s disease, and synthetized from L-DOPA, behaves as a peripheral catechol-*O*-methyltransferase inhibitor (COMTI). Therefore, it may be a promising drug candidate for the development of COMT inhibitors useful in the treatment of Parkinson’s disease [11]. 1,2,3,4-Tetrahydroisoquinoline-1-carboxylic acid (**C**) (Figure 1) can efficiently inhibit in vitro the activity of NDM-1 (New Delhi metallo-β-lactamase), which is widespread in many bacteria and is able to hydrolyze almost all β-lactam antibiotics [12].

Considerable effort has been devoted to asymmetric synthesis of tetrahydroisoquinolines [13,14]. Among the variety of novel methods of the synthesizing isoquinoline core, an important role is played by the modification of traditional methods such as Pictet–Spengler cyclization, Bischler–Napieralski cyclization/reduction, or Pomeranz–Fritsch–Bobbitt cyclization. 

In this paper, we present examples published in last two decades of the application of the Pomeranz–Fritsch–Bobbitt process in the synthesis of natural products and their derivatives, among them the novel synthesis of (–)-6,7-dimethoxy-1,2,3,4-tetrahydroisoquinoline-1-carboxylic acid (**1**) (Figure 2). This compound and its enantiomer are related to simple isoquinoline alkaloids [15] and important arylglycine derivatives [16], so considerable efforts have been devoted to their stereoselective preparation. Fülöp and colleagues [17] reported the first high-yield preparation of both enantiomers of 6,7-dimethoxy-1,2,3,4-tetrahydroisoquinoline-1-carboxylic acid (**1** and *ent*-**1**), based on chemoenzymatic kinetic resolution of its racemic ethyl ester. Wu and colleagues [8,9] presented a method for efficient deracemization of 1,2,3,4-tetrahydroisoquinoline carboxylic acids (**1**) using D-amino acid oxidase from *Fusarium solani* (*Fs*DAAO) as a biocatalyst. We performed diastereoselective synthesis of (+)-6,7-dimethoxy-1,2,3,4-tetrahydroisoquinoline-1-carboxylic acid (*ent*-**1**) [18], applying Pomeranz–Fritsch–Bobbitt [19,20,21,22,23,24,25] cyclization coupled with the Petasis reaction [26,27,28,29,30].

Caesar Pomeranz [15,31] and Paul Fritsch [15,19] in 1893 independently reported a new method of isoquinoline synthesis from benzaldehyde and 2,2-diethoxyethylamine. Since that time, this reaction has been extensively studied. In 1948, Schlittler and Müller [32] modified the reaction by using benzyl amines and glyoxal hemiacetal as the starting compounds. Later, Bobbitt [20,21,22,23,24,25] reported synthesis of 1,2,3,4-tetrahydroisoquinolines by hydrogenation of the imine intermediate in situ to the aminoacetal, allowing preparation of 1-, 4-, and *N*-substituted isoquinolines. At the same time, Jackson [33] described the dehydrogenation of 1,2-dihydroisoquinolines via *N*-tosyl derivatives to a fully aromatic system.

The term “Pomeranz–Fritsch–Bobbitt synthesis” refers to a variety of synthetic strategies in which the nitrogen-containing heterocyclic ring B is built including formation of C4-C4a bond of benzylamines, bearing a two-carbon C-C chain at the nitrogen atom with a good alkoxy leaving group (Figure 1) [13,14].

In recent decades, research using Pomeranz–Fritsch–Bobbitt methodology has principally focused on the asymmetric synthesis of 1,2,3,4-tetrahydroisoquinolines as reported in literature [13,14].

Badia and colleagues [34] used a Pomeranz–Fritsch–Bobbitt process in the diastereoselective synthesis of (*S*)*-*laudanosine (**6**), a benzylisoquinoline alkaloid, employing (*S*)-phenylglycinol-derived imine **2** and Grignard reagent **3** as substrates (Figure 2). In the obtained addition product, the chiral auxiliary was removed by catalytic hydrogenation to give primary amine **4**. *N*-Alkylation of compound **4** with 2,2-diethoxyethylbromide led to aminoacetal **5**. *N*-methylation followed by Pomeranz–Fritsch cyclization and ionic hydrogenation converted it into (*S*)-(+)*-*laudanosine (**6**) in high yield with 94% e.e.

Both enantiomers of protoberberine alkaloid, (*S*)- and (*R*)-*O*-methylbharatamine (**12** and *ent*-**12**) [35], were synthesized by our group using diastereoselective Pomeranz–Fritsch–Bobbitt cyclization. In this approach, chiral *o*-toluamides (*S*)-**8** and (*R*)-**8** derived from (*S*)- and (*R*)-phenylalaninol, respectively, were prepared incorporating oxazolidine moiety and added to achiral imine acetal **7** using the lateral metallation methodology [36] (Figure 3). This resulted in the formation of two types of compounds: the addition/partial cyclization products isoquinolones **9** and the acyclic addition products **10**. Compounds **10** were easily converted into isoquinolones **9** under the action of *n*-BuLi. Reduction of amides (*S*)-**9** and (*R*)-**9**, followed by cyclization/reduction of amino acetals (*S*)-**11** and (*R*)-**11** completed the synthesis of (*S*)- and (*R*)-*O*-methylbharatamine (**12** and *ent*-**12**) obtained in 32% and 37% overall yield and 88% e.e. and 73% e.e., respectively.

A complementary diastereoselective synthesis of (*S*)-*O*-methylbharatamine (**12**), starting with chiral sulfinaldimine and achiral *o*-toluamide has also been invented in our laboratory [37] (Figure 4). The addition of laterally lithiated *N,N*-diethyl *o*-toluamide **14** to (*S*)-*N*-*tert*-butane sulfinimine **13** gave addition product **15**. Removal of the *N*-sulfinyl auxiliary led to amine **16,** which upon treatment with *n*-BuLi cyclized easily into isoquinolone **17**. Isoquinolone **17** was then reduced and *N*-alkylated with bromoacetaldehyde diethyl acetal to aminoacetal **18,** which when subjected to cyclization/hydrogenolysis afforded (*S*)-*O*-methylbharatamine (**12**) in 24% overall yield with 88% e.e.

Enders and Boudou [38] described a stereoselective synthesis of both (*S*)- and (*R*)-enantiomers of tetrahydropalmatine (**23** and *ent*-**23**), a protoberberine alkaloid, based on the addition of laterally lithiated toluamide **20** to SAMP or RAMP [(*S*)- or (*R*)-1-amino-2-(methoxymethyl)pyrrolidinyl] hydrazones **19**, respectively. The intermediate addition/cyclization products **21** were obtained in high diastereomeric purity (d.e. ≥ 96%) and 55% yield (Figure 5). Cleavage of the N-N bond of the chiral auxiliary in compound **21** and reduction of the lactam carbonyl followed by introduction of 2,2-diethoxyethyl substituent to nitrogen atom yielded amine acetals (*S*)-**22** or (*R*)-**22**. The Pomeranz–Fritsch–Bobbitt cyclization of compounds (*S*)-**22** or (*R*)-**22** led to (*S*)- and (*R*)-tetrahydropalmatine (**23** and *ent*-**23**), with 98% e.e. (both enantiomers) and overall yields of 17% and 9%, respectively.

Diastereoselective total synthesis of both enantiomeric (*S*)-salsolidine (**28**) [39] and (*R*)-salsolidine (*ent*-**28**) [40] using Pomeranz–Fritsch–Bobbitt cyclization for the construction of the tetrahydroisoquinoline ring system was also performed in our laboratory (Figure 6). Chiral *N*-*tert*-butanesulfinylimines *ent*-**13** and **24** were used as substrates. Chiral α-benzylamines (*S*,*R*)-**25** and (*R*,*R*)-**25** were prepared using two parallel reaction pathways. Addition of methylmagnesium bromide to aldimine *ent*-**13** provided α-methylamine (*S*,*R*)-**25**. The diastereomeric amine (*R*,*R*)-**25** was prepared by hydride reduction of ketimine **24**. Removal of the chiral auxiliary in diastereoisomers of **25** led to enantiomeric amines (*S*)-**26** and (*R*)-**26** isolated as hydrochloride salts. *N*-alkylation of amines (*S*)-**26** and (*R*)-**26** with 2,2-diethoxyethyl bromide led to enantiomeric acetals (*S*)-**27** and (*R*)-**27**. Final cyclization/hydrogenolysis of (*S*)-**27** gave (*S*)-salsolidine (**28**) in 24% overall yield with 98% e.e., while cyclization/hydrogenolysis of (*R*)-**27** afforded (*R*)-salsolidine (*ent*-**28**) in 20% overall yield with 96% e.e.

Enantioselective synthesis of (*S*)-salsolidine (**28**) was part of the work carried out in our laboratory [41]. Additions of methyllithium to the imine **7**, carried out in the presence of several oxazoline chiral ligands, type **29**, led to aminoacetal **30** of known (*S*) configuration. The yields differed from 44–92% and enantioselectivity was up to 76% e.e., depending on the type of oxazoline **29** used (Figure 7). The Pomeranz–Fritsch–Bobbitt cyclization of aminoacetal **30** led to (*S*)-(–)-salsolidine (**28**) [42].

A modified procedure for the classical Pomeranz–Fritsch protocol was presented by the Lumb Group [43]. They showed that the strong acids and elevated temperatures used in the Pomeranz–Fritsch cyclization step can be replaced by a combination of silyl triflate and a sterically encumbered pyridine base [44], which allows acetal activation under milder, more chemoselective conditions. This modification tolerates acid-sensitive functional groups in substrates and facilitates the synthesis of diverse 1,2-dihydroisoquinoline derivatives of type **32**. From acetals of type 31. Further functionalization of these compounds led to reduced 1,2,3,4-tetrahydroisoquinolines of type **33** (Figure 8).

The same Pomeranz–Fritsch cyclization conditions were employed in the asymmetric synthesis of (*S*)-cularine (**36**) [45]. Aminoacetal **34** was treated with an excess of trimethylsilyl trifluoromethanesulfonate (TMSOTf) and 2,6-lutidine in DCM to give 1,2-dihydroisoquinoline derivative **35** in 83% yield, and no loss of enantioselectivity was observed. Several transformations including, among others, hydrogenation, oxidation, coupling with 4-nitrophenol, cyclization to dibenzodioxepin, and final metylation of catechol led to (*S*)-cularine (**36**) in 15% overall yield and with 97% o.p. (Figure 8).

In our research, the Pomeranz–Fritsch–Bobbitt [19,20] cyclization was coupled with the Petasis reaction [26] to afford C-1 substituted tetrahydroisoquinoline derivatives: C-1 carboxylic acids [46], simple isoquinoline alkaloids [47], and 7,12-dihydro-6,12-methanodibenzo[*c*,*f*]azocine-5-carboxylic acids [48].

We performed a diastereoselective synthesis of (+)-6,7-dimethoxy-1,2,3,4-tetrahydroisoquinoline-1-carboxylic acid (*ent*-**1**) with 90% e.e. [18] using chiral aminoacetaldehyde acetals type **37** incorporating the chiral amines (*S*)-(–)-phenylethylamine, (*S*)-(–)-α-naphthylethylamine, (*S*)-(+)-tetrahydronaphthyl-1-amine, and (*R*)-(–)-1-indanamine as chiral inductors of the Petasis step. Reaction between 3,4-dimethoxyphenyl boronic acid **38**, glyoxylic acid monohydrate **39,** and aminoacetaldehyde dimethyl acetals of type **37** afforded amino acids **40,** being coupling acetal products in high yield as a mixture of diastereomers (from 79:21 to 56:44 d.r.), dependent on the chiral inductor applied (Figure 9). Removal of the *N-*chiral auxiliary from intermediate acetal **40** led to *N*-deprotected amino acid **41**, obtained in enantiomerically enriched form after recrystallization of the crude reaction product. Cyclization/hydrogenolysis completed the synthesis, supplying the target dextrorotatory amino acid *ent*-**1** with 90% e.e.

In our earlier experiments, we found that in the reactions in which *N*-substituted 2-phenylglycinol was applied, the initially formed Petasis product, the corresponded hydroxy amino acids, underwent direct cyclization, leading to 3,5-disubstituted 1,4-oxazin-2-ones derivatives.

The reaction of (*R*)-*N*-benzyl-2-phenylglycinol (**42**) with 3,4-dimethoxyphenylboronic acid (**38**) and glyoxylic acid monohydrate (**39**) led to two diastereomeric oxazinones **43** and **44** in 98% yield with 58:42 d.r., which were easily separated by column chromatography (Figure 10) [49]. Absolute configuration of the major isomer **43** was established by X-ray crystal analysis to be *trans*-(3*R*,5*R*).

## 2. Results

In this paper we describe synthesis of enantiomeric (–)-6,7-dimethoxy-1,2,3,4-tetrahydroisoquinoline-1-carboxylic acid (**1**), applying the Petasis/Pomeranz–Fritsch–Bobbitt approach in which chiral oxazinone bearing acetal moiety was the key intermediate. Our retrosynthetic approach for the synthesis of 6,7-dimethoxy-1,2,3,4-tetrahydroisoquinoline-1-carboxylic acid is shown in Figure 11. In this synthesis, the concept was to involve chiral aminoacetaldehyde acetal derived from (*R*)-phenylglycinol in the Petasis reaction with boronic acid and glyoxylic acid to afford a rigid chiral oxazinone derivative, which was further transformed into the substrate for Pomeranz–Fritch–Bobbitt cyclization leading to 6,7-dimethoxy-1,2,3,4-tetrahydroisoquinoline-1-carboxylic acid.

Chiral aminoacetal **47** incorporating (*R*)-phenylglycinol moiety was synthesized in the reaction of (*R*)-phenylglycinol (**45**) with excess of 2-bromo-1,1-diethoxyethane (**46)** carried out in dry DMF in the presence of anhydrous K_2_CO_3_ at 110 °C. After purification by column chromatography, the product was isolated in 67% yield. When the same reaction was carried with an equimolar ratio of (*R*)-phenylglycinol (**45**) and **46**, low conversion was observed as indicated by TLC. The next step of the synthesis was the Petasis reaction with boronic acid **38**, glyoxylic acid monohydrate (**39),** and aminoacetaldehyde acetal **47** (Figure 12).

The substrates **38**, **39**, and **47** were stirred in DCM at room temperature for 78 h, then the inorganic solid was removed by filtration. The filtrate was evaporated in vacuo to give an oily residue consisting of diastereomeric oxazin-2-ones **48** and **49** with 3:1 d.r. It was necessary to carry out this reaction for at least 78 h, because at shorter reaction times substantial amounts of unreacted substrates **38** and **47** were present in the reaction mixture as well as two unidentified products, considerably more polar than **48** and **49**, probably being their noncyclized precursors (not shown). 

The diastereomeric composition of oxazin-2-ones was established by ^1^H NMR spectroscopic analysis of the crude reaction mixture. The diastereomeric ratio of 3:1 for compounds **48** and **49** was deduced by comparing the integration values for the H-3 oxazin-2-one ring proton signals at 4.91 and 5.05 ppm, respectively, and integration values for the methyl groups of the acetal moiety signals at 1.21 and 1.12 ppm. 

After purification of the crude product by column chromatography, pure major diastereomer **48** (51%) and minor diastereomer **49** (16%) were isolated as colourless oils.

The major diastereomer **48** was the key intermediate for the synthesis of **1**. 

Debenzylation of oxazinone **48** to aminoacetal **50** (racemic **50** ref. [46]) was performed by hydrogenolysis conducted under atmospheric pressure (using a reaction balloon filled with hydrogen gas) in the presence of palladium hydroxide on charcoal (Pearlman’s catalyst) (Figure 13). The crude product was pure enough for the next reaction step, at approximately 90% as determined by ^1^H NMR, but contained some minor unidentified impurities. Therefore, it was purified by column chromatography to give the pure (*R*)-(–)-*N*-(2,2-diethoxyethyl)-3,4-dimethoxyphenylglycine (**50**) in 66% yield as a yellowish solid.

The Pomeranz–Fritsch–Bobbitt cyclization of aminoacetal **50** was carried out in 20% HCl at rt for 72 h under argon in darkness. The reaction mixture was then hydrogenated for 24 h under a hydrogen atmosphere (H_2_ balloon) in the presence of 10% Pd-C catalyst. The catalyst was removed by filtration and the filtrate was evaporated in vacuo to obtain the crude hydrochloride salt of **1**. It was treated with 25% ammonium hydroxide diluted with chloroform-methanol to pH ≈ 6 at rt. Evaporation of the solvents and trituration of the crude product with a mixture of 96% ethanol/water afforded pure, crystalline (*R*)-(–)-6,7-dimethoxy-1,2,3,4-tetrahydroisoquinoline-1-carboxylic acid (**1**) showing [*α*]_D_ –59.5 (*c* = 0.39, H_2_O) (*rac*-**1**: ref. [46], *ent*-**1**: ref. [18]). A sample of the levorotatory enantiomer (99% e.e.), as described by Fülöp and colleagues, [17] was characterized by [α]_D_ –63 (c = 0.30, H_2_O).

## 3. Conclusions

The diastereoselective total synthesis of (–)-6,7-dimethoxy-1,2,3,4-tetrahydroisoquinoline-1-carboxylic acid (**1**) was designed and achieved in three simple steps by applying two synthetic methods, namely, the Petasis synthesis of amino acids and the Pomeranz–Fritsch–Bobbitt synthesis of tetrahydroisoquinoline derivatives. The stereoselectivity of the synthesis was directed by (*R*)-*N*-(2,2-diethoxyethyl)-2-phenylglycinol (**47**), which was used as the amine component of the Petasis reaction. The Petasis products, two diastereomeric oxazin-2-ones **48** and **49**, were formed with 3:1 d.r., (by ^1^H NMR) and were separated and isolated in pure form. The major diastereomer **48** was the key intermediate in the next part of the synthesis. Debenzylation of this compound led to aminoacetal **50** and the next Pomeranz–Fritsch–Bobbitt step afforded the target (–)-6,7-dimethoxy-1,2,3,4-tetrahydroisoquinoline-1-carboxylic acid (**1**).

## 4. Experimental Section

### 4.1. General

Melting points were determined by using open glass capillaries in a Büchi melting point B-545 apparatus and are reported uncorrected. IR spectra were recorded on a Bruker FT-IR IFS 113V spectrophotometer or Jasco FT-IR 4600 spectrophotometer with ATR PRO ONE using a diamond crystal. The ^1^H and ^13^C NMR spectra were recorded on a Bruker ASCEND 400 spectrometer. NMR spectra are reported in parts per million (ppm) and were measured relative to the signals for residual solvent peak CDCl_3_ or DMSO-d_6_, or using tetramethylsilane as an internal reference. Mass spectra (EI) were measured using an AMD402 spectrometer. High-resolution mass spectra (HRMS) were measured using an Impact HD (Bruker Daltonics, Bremen, Germany) spectrometer. High-resolution ESI-MS spectra were recorded on a quadrupole time-of-flight mass spectrometer (QTOF, Impact HD, Bruker Daltonics, Bremen, Germany) in positive mode. Merck DC-Alufolien Kieselgel 60_254_ were used for TLC, and silica gel (100–200 mesh ASTM) for column chromatography. Optical rotation was measured using a Perkin–Elmer polarimeter 242B at 20 °C. All reagents and solvents were purchased from commercial suppliers and used as received. 

### 4.2. (R)-(–)-N-(2,2-Diethoxyethyl)-2-phenylglycinol *(**47**)*

(*R*)-phenylglycinol (**45**) (798 mg, 5.82 mmol), anhydrous K_2_CO_3_ (1465 mg, 10.6 mmol) and DMF (6 mL) followed by 2-bromo-1,1-diethoxyethane (**46**) (1380 mg, 1.05 mL, 7 mmol) were added to a round-bottomed flask equipped with a magnetic stirrer and reflux condenser with drying tube attached. The reaction mixture was heated at 110 °C for 24 h. During that time, an additional quantity of **46** (1380 mg, 1.05 mL, 7 mmol) was added. After completion of the reaction, the mixture was poured onto ice and after reaching rt was extracted four times with ethyl ether. The organic extracts were dried over anhydrous MgSO_4_ and the solvent was evaporated to afford crude **47**, which after purification by silica gel column chromatography (DCM-MeOH 100:0 to 98:2 *v*/*v*) yielded aminoacetal **47** as a solidifying oil (988 mg, 67%).

[α]_D_—56.8 (*c* 0.95, DCM).

^1^H NMR (400 MHz, CDCl_3_) δ 7.30–7.18 (m, 5H), 4.51 (t, *J* = 5.5 Hz, 1H), 3.72 (dd, *J* = 8.7, 4.4 Hz, 1H), 3.67–3.49 (m, 4H), 4.46–3.37 (m, 2H), 2.66–2.54 (m, 2H), 2.48 (broad s, it disappears after treatment with D_2_O, 2H), 1.13 (t, *J* = 7.0 Hz, 3H), 1.12 (t, *J* = 7.0 Hz, 3H).

^13^C NMR (100 MHz, CDCl_3_) δ 140.0, 128.6, 127.7, 127.2, 101.8, 66.5, 64.6, 62.5, 62.1, 49.5, 15.3.

IR (neat): *ṽ* = 3291, 3132, 2979, 2834, 1360, 1058.

EI MS: *m*/*z* (%): 253 (M^+^, 5), 221 (100), 175 (41), 161 (12), 120 (19).

HRMS (ESI) calculated for C_14_H_24_NO_3_ [M + H]^+^ 254.1756; found at 254.1752.

### 4.3. Synthesis of (3R,5R) and (3S,5R)-4-(2,2-diethoxyethyl)-3-(3,4-dimethoxyphenyl)-5-phenyl-1,4-oxazin-2-ones *(**48**)* and *(**49**)*

(*R*)-*N*-(2,2-diethoxyethyl)-2-phenylglycinol (**47**) (927 mg, 3.66 mmol) and glyoxylic acid monohydrate (**39**) (337 mg, 3.66 mmol) in DCM (18 mL) were stirred in a round-bottomed flask at room temperature for 5 min. Then, 3,4-dimethoxyphenyl boronic acid (**38**) (666 mg, 3.66 mmol) was added and the mixture was stirred at room temperature for 78 h. After this time, the inorganic solid was removed by filtration and the solvent was evaporated in vacuo to give an oily residue (1640 mg) with 3:1 d.r. (by NMR). The crude product was purified by silica gel column chromatography (hexane:ethyl acetate 75:25 *v*/*v*) to afford pure major isomer **48** (803 mg, 51%) and minor isomer **49** (254 mg, 16%) as colourless oils.

*(3R,5R)-(–)-4-(2,2-Diethoxyethyl)-3-(3,4-dimethoxyphenyl)-5-phenyl-1,4-oxazin-2-one* (**48**) *(major isomer)*

[α]_D_—137.8 (*c* 0.97, MeOH). 

^1^H NMR (400 MHz, CDCl_3_) δ 7.40–7.31 (m, 2H), 7.10–7.08 (m, 2H), 6.89–6.86 (m, 1H), 4.91 (s, 1H), 4.82 (dd, *J* = 11.1, 4.0 Hz, 1H), 4.58 (dd, *J* = 11.1, 5.4 Hz, 1H), 4.52 (dd, *J* = 5.3, 4.0 Hz, 1H), 4.36 (dd, *J* = 6.4, 4.2 Hz, 1H), 3.91 (s, 3H), 3.89 (s, 3H), 3.65–3.58 (m, 1H), 3.46–3.38 (m, 2H), 3.29–3.21 (m, 1H), 2.78 (dd, *J* = 14.2, 6.4 Hz, 1H), 2.52 (dd, *J* = 14.1, 4.2 Hz, 1H), 1.21 (t, *J* = 7.1 Hz, 3H), 1.05 (t, *J* = 7.0 Hz, 3H).

^13^C NMR (100 MHz, CDCl_3_) δ 169.6, 149.0, 148.9, 137.3, 129.7, 128.8, 128.2, 128.1, 120.5, 111.4, 111.0, 103.0, 71.8, 66.1, 62.8, 62.0, 56.8, 55.9, 51.8, 15.3, 15.2.

IR (KBr): *ṽ* = 2973, 2836, 1743, 1514, 1464, 1262.

EI MS: *m*/*z* (%): 429 (M^+^, 3), 325 (14), 178 (12), 151 (31), 103 (100).

HRMS calculated for C_24_H_31_NO_6_: 429.21515; found at 429.21400.

*(3S,5R)-(+)-4-(2,2-Diethoxyethyl)-3-(3,4-dimethoxyphenyl)-5-phenyl-1,4-oxazin-2-one* (**49**) *(minor isomer)*

[α]_D_ + 60.2 (*c* 0.52, MeOH). 

^1^H NMR (400 MHz, CDCl_3_) δ 7.50–7.53 (m, 2H), 7.42–7.32 (m, 4H), 7.28–7.25 (m, 1H), 6.91 (d, *J* = 8.3 Hz, 1H), 5.05 (t, *J* = 0.8 Hz, 1H), 4.49 (t, *J* = 5.3 Hz, 1H), 4.28 (dd, *J* = 10.7, 2.8 Hz, 1H), 4.22 (t, *J* = 10.7 Hz, 1H), 4.12 (dd, *J* = 10.6, 2.8 Hz, 1H), 3.94 (s, 3H), 3.90 (s, 3H), 3.53–3.42 (m, 2H), 3.36–3.28 (m, 1H), 3.17–3.24 (m, 1H), 2.85 (dd, *J* = 5.1, 14.3 Hz, 1H), 2.80 (dd, *J* = 5.5, 14.4 Hz, 1H), 1.13 (t, *J* = 7.0 Hz, 3H), 1.02 (t, *J* = 7.1 Hz, 3H).

^13^C NMR (100 MHz, CDCl_3_) δ 170.3, 149.0, 148.6, 137.5, 131.3, 128.8, 128.5, 128.0, 118.9, 111.0, 110.3, 101.1, 70.8, 66.3, 63.0, 62.7, 61.8, 55.9, 55.9, 55.5, 15.3, 15.1.

IR (KBr): *ṽ* = 2974, 2898, 1748, 1513, 1464, 1262.

EI MS: *m*/*z* (%): 429 (M^+^, 13), 326 (18), 311 (17), 220 (16), 178 (14), 151 (29), 103 (100).

HRMS calculated for C_24_H_31_NO_6_: 429.21515; found at 429.21290.

### 4.4. (R)-(–)-N-(2,2-Diethoxyethyl)-3,4-dimethoxyphenylglycine *(**50**)*

The major Petasis reaction product **48** (563 mg, 1.31 mmol) was dissolved in ethanol (HPLC grade) (10 mL) in a two-neck round-bottomed flask. The Pearlman catalyst (310 mg) was added to the solution and the reaction mixture was hydrogenated under an atmosphere of H_2_ (hydrogen from a balloon) for 24 h at ambient temperature and pressure. Then, the catalyst was removed by filtration, the filter cake was washed several times with ethanol, and the solvent was evaporated to afford 348 mg (81%) of crude **50**, which was further purified by silica gel column chromatography (DCM:MeOH 10:4 *v*/*v*) to give pure **50** (283 mg, 66%) as a yellowish solid.

m.p. 132–134 °C 

[α]_D_—75.8 (*c* 0.65, MeOH). 

^1^H NMR (400 MHz, CDCl_3_) δ 7.02–7.00 (m, 2H), 6.81 (d, *J* = 8.1 Hz, 1H), 4.71–4.69 (m, 2H), 3.84 (s, 3H), 3.83 (s, 3H), 3.57–3.51 (m, 2H), 3.47–3.42 (m, 1H), 3.39–3.34 (m, 1H), 2.95 (dd, *J* = 12.5, 4.4 Hz, 1H), 2.65 (dd, *J* = 12.5, 6.3 Hz, 1H), 1.13 (t, *J* = 7.1 Hz, 3H), 1.08 (t, *J* = 7.0 Hz, 3H).

^13^C NMR (100 MHz, CDCl_3_) δ 171.8, 149.5, 149.3, 125.9, 121.5, 111.3, 98.8, 64.8, 63.5, 63.0, 55.9, 55.9, 45.9, 15.1, 15.1.

All other spectroscopic data were in accordance with the literature and with those of the racemic compound [46].

### 4.5. (R)-(–)-6,7-Dimethoxy-1,2,3,4-tetrahydroisoquinoline-1-carboxylic acid *(**1**)*

Aminoacetal **50** (265 mg, 0.81 mmol) was dissolved in 20% hydrochloric acid (6 mL) in a two-neck round-bottomed flask, and the resulting solution was stirred at room temperature for 72 h under argon in darkness. Then, the mixture was hydrogenated (hydrogen from a balloon) in the presence of 10% Pd/C (380 mg) for 24 h. The catalyst was removed by filtration, the filter cake was washed several times with water, and the filtrate was concentrated under high vacuum to deposit the crude hydrochloride salt of the isoquinoline carboxylic acid **1·HCl** (185 mg) as a yellowish foam. The **1·HCl** was treated with ammonium hydroxide (25%) diluted with chloroform/methanol (NH_4_OH/CHCl_3_/CH_3_OH, 0.5:45:4.5 *v*/*v*/*v*) to pH ≈ 6 at room temp. for 15 min. Evaporation of solvents and trituration of the crude product with 96% ethanol/water (2.5 mL, 2:0.5 *v*/*v*) afforded pure, crystalline (*R*)-(–)-6,7-dimethoxy-1,2,3,4-tetrahydroisoquinoline-1-carboxylic acid (1, 112 mg, 58%). 

[*α*]_D_–59.5 (*c* = 0.39, H_2_O).

^1^H NMR (400 MHz, D_2_O) δ 2.99 (t, *J* = 6.3 Hz, 2 H), 3.44 (td, *J* = 6.4, 12.7 Hz, 1 H), 3.58 (td, *J* = 6.3, 12.9 Hz, 1 H), 3.84 (s, 3 H), 3.87 (s, 3 H), 4.86 (s, 1 H), 6.85 (s, 1 H), 7.15 (s, 1 H).

^13^C NMR (100 MHz, D_2_O) δ 27.0, 42.7, 58.5, 58.5, 61.1, 113.4, 114.3, 123.1, 127.3, 149.8, 150.9, 175.0.

All other spectroscopic data were in accordance with the literature [*ent*-1: ref. [18], *rac*-1: ref. [46]].

## Data Availability

Supporting data is provided as Appendix A.

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
