# Peer review of "Diastereoselective Synthesis of (–)-6,7-Dimethoxy-1,2,3,4-tetrahydroisoquinoline-1-carboxylic Acid via Morpholinone Derivatives"

_molecules, 2023, doi:10.3390/molecules28073200_

Round 1

Reviewer 1 Report

This paper describes a diastereoselective synthesis of chiral isoquinoline derivative 1. Authors have continuously reported this type transformation using diastereoselective transformation strategies and this reviewer recognized the present work is the simple extension and combination of the previous works ref 10 and 32. This manuscript is submitted as an article, but the first half is written in the review style. First of all, I felt uncomfortable with that. If it is considered as an article, the actual content of this article would correspond to those after lines 192, which only describes the separation of diastereomers with 3:1 selectivity and their derivation to compound 1. The content of the article did not seem to have any novelty. Despite the interesting product structures from a viewpoint of medicinal chemistry, to this reviewer, these results are still short of significance for this journal. I thus considered there is no remarkable novelty to be published in Molecules (IF 4.927).

Reviewer 2 Report

The manuscript entitled “Diastereoselective synthesis of(-)-6,7-dimethoxy-1,2,3,4-tetrahydroisoquinoline-1-carboxylic acid via morpholinone derivtives” from Chrzanowska et al involved the interesting preparation of compounds containing the isoquinoline core through two synthetic methodologies.

Additionally, the authors reported a complete review analysis of both methods and their importance in the organic synthesis field of research.

The introduction is according to the theme of the manuscript, and it has revised bibliographical references to support the research the authors have done.  Also, it contained the previous related results to support the selection of the reported organic moieties.

Furthermore, the manuscript is clear, organize, and focused on the topic.

This reviewer suggests including the potential applications and the uses of these kind of molecules in order to engage the potential readers.

In addition, it has an organize, complete, and suitable methodologies to produce the desire organic compounds (with complete structural characterization).

The syntheses of the proposed compounds were performed via the Petasis/Pomeranz-Fritsch-Bobbitt approaches, and the authors followed the stablished methods for these purposes.

As a suggestion, it could be important to specify why the authors didn’t explore additional methodologies such as the MW utilization or ultrasound energy.

Also, the results are organized, and they are analyzed and discussed according to the relevant obtained results.

Taking to account the results there are some questions I would like to point out:

Moreover, I encourage the authors to check some mistakes (in yellow, pdf file attached) such as:

-     Please remember that the unit and the number (e.g. 2 N – 50 °C – 1 h) should have a blank space between them.

-     Line 232: in vacuo should be in italics.

Author contributions: please follow the journal rules to complete this part of the manuscript. (https://www.mdpi.com/ethics#_bookmark3 )  

The following statements should be used: Conceptualization, X.X. and Y.Y.; methodology, X.X.; software, X.X.; validation, X.X., Y.Y. and Z.Z.; formal analysis, X.X.; investigation, X.X.; resources, X.X.; data curation, X.X.; writing—original draft preparation, X.X.; writing—review and editing, X.X.; visualization, X.X.; supervision, X.X.; project administration, X.X.; funding acquisition, Y.Y. All authors have read and agreed to the published version of the manuscript.

Furthermore, I encourage the authors to include a table summarizing all the information reported in the present manuscript.

Moreover, the supplementary information file should be included to review it in order to check the data belonging to the spectroscopic analyses, thanks. (To clarify and confirm the reported results)

Finally, I would like to invite the authors to add the abbreviation list of words at the end of this manuscript.

Reviewer 3 Report

Dear Authors,

Quality of the work is interesting and I enjoyed reading this work but I like to see a little modification in your manuscript.

I like to see some sentences about heterocyclic chemistry and their bioactivity in the introduction part. besides, it makes the manuscript more amazing when you bring some sentences about different nitrogen-based heterocyclic bioactive compounds. in parallel, I think you have to more updated references because many of your references is tooooooo OLD. Here you can find some updated references related to nitrogen based heterocyclic bioactive compounds for using in your manuscript. 

1.Pyrazole Derivatives Induce Apoptosis via ROS Generation in the Triple Negative Breast Cancer Cells, MDA-MB468. doi: 10.31557/APJCP.2021.22.7.2079

2.Synthesis and biological evaluation of 2-(2-methyl-1H-pyrrol-3-yl)-2-oxo-N-(pyridine-3-yl) acetamide derivatives: in vitro α-glucosidase inhibition, and kinetic and molecular docking study. 

https://doi.org/10.1007/s11696-019-00999-0

3. Design, synthesis and antitumour evaluation of pyrrolo[1,2-f]-phenanthridine and dibenzo[f,h]pyrrolo[1,2-b]isoquinoline derivatives. https://doi.org/10.1016/j.ejmech.2020.112516

4. 2,4-Disubstituted Quinazoline Derivatives Act as Inducers of Tubulin Polymerization: Synthesis and Cytotoxicity. https://doi.org/10.2174/1871520619666190314125254

5. Synthesis and Cytotoxic Activity Study of Novel 2-(Aryldiazenyl)-3-methyl-1H-benzo[g]indole Derivatives. https://doi.org/10.3390/molecules26144240

6. Design, synthesis and biological evaluation of 3-arylisoquinoline derivatives as topoisomerase I and II dual inhibitors for the therapy of liver cancer. https://doi.org/10.1016/j.ejmech.2022.114376

Round 2

Reviewer 1 Report

I have read and understand the authors' response. I believe it is acceceptable.